# Nutritional quality of retail food purchases is not associated with participation in the Supplemental Nutrition Assistance Program for nutrition-oriented households

Yu Chen[1], Biing-Hwan Lin[2], Lisa Mancino[2], Michele Ver Ploeg[3], Chen Zhen[4]*

**1** Formerly with Department of Agricultural and Applied Economics, University of Georgia, Athens, GA, United States of America, **2** Formerly with Economic Research Service, U.S. Department of Agriculture, Washington, DC, United States of America, **3** Food and Health Policy Institute, Milken Institute School of Public Health, The George Washington University, Washington, DC, United States of America, **4** Department of Agricultural and Applied Economics, University of Georgia, Athens, GA, United States of America

* czhen@uga.edu

**Data Availability Statement:** The FoodAPS data used in this study is publicly available for download from the US Department of Agriculture (https://

## Abstract

The Supplemental Nutrition Assistance Program (SNAP) provides millions of low-income Americans food benefits and other forms of nutrition assistance. Evidence indicates that SNAP reduces food insecurity. However, there is a concern that the food benefit may increase the demand for less healthy foods more than healthier foods, thereby reducing the overall nutritional quality of the participant's food basket. This paper aims to examine the association of SNAP participation with the nutritional quality of food-at-home purchases of low-income households and to investigate the potential heterogeneity among consumers with different levels of nutrition attitude. This analysis used food purchase data from the USDA National Household Food Acquisition and Purchase Survey (FoodAPS). Our study sample included 2,218 low-income households, of which 1,184 are SNAP participants, and 1,034 are income-eligible nonparticipants. Multivariate regressions were performed to explore the SNAP-nutritional quality association. A household's nutrition attitude was measured using its response to a question on whether the household searched for nutrition information online in the last 2 months. Households that affirmed they had an online nutrition search were treated as nutrition-oriented households (21.2% of the low-income sample), and households that did not were considered less nutrition-oriented households (78.8%). For robustness, we also created an alternative nutrition attitude measure based on reported use of the nutrition facts label. We found that among less nutrition-oriented households, SNAP participants had a statistically significant 0.097 points ($p = 0.018$) lower Guiding Stars rating than low-income nonparticipants. However, there was no significant SNAP-nutritional quality association among nutrition-oriented households. In conclusion, SNAP participation was associated with lower nutritional quality of food purchases among less nutrition-oriented households, but not among nutrition-oriented households. The results suggest that the intended nutritional benefits of restrictions on purchases of healthy foods may not reach the subgroup of nutrition-oriented SNAP participants.

www.ers.usda.gov/data-products/foodaps-national-household-food-acquisition-and-purchase-survey/).

**Funding:** CZ is the author who received the award. This research was supported by the intramural research program of the U.S. Department of Agriculture, Economic Research Service and cooperative agreement 58-4000-6-0052. The findings and conclusions in this publication are those of the author(s) and should not be construed to represent any official USDA or U.S. Government determination or policy. The funder website is https://www.ers.usda.gov/. The funders had no role in study design, data analysis, decision to publish, or preparation of the manuscript.

**Competing interests:** The authors have declared that no competing interests exist.

## Introduction

Suboptimal diet is related to increased risks of obesity and non-communicable diseases such as heart disease, diabetes and certain cancers [1]. In 2017, suboptimal diets were associated with 11 million deaths and a loss of 255 million disability-adjusted life-years globally [2]. There is evidence that low-income populations purchase and consume foods of lower nutritional quality than their higher-income counterparts [3].

The Supplemental Nutrition Assistance Program (SNAP) is the largest U.S. domestic hunger safety net program, serving 35.7 million eligible low-income Americans in fiscal year 2019 with $55.6 billion in food benefits. To be eligible for SNAP, households must meet a variety of income and resources tests. For example, households must have net income at or below 100% of the Federal poverty thresholds (FPL), and households without an elderly or disabled member must have monthly gross income at or below 130% FPL. A household may be categorically eligible for SNAP if it is eligible for certain other assistance programs [4]. About 85 percent of eligible individuals participated in SNAP in 2016 [4]. Participation is voluntary and the decision depends on a number of policy, environmental, social, and economic factors. Previous studies suggested that eligible working-poor families and low-income elderly had low participation rates [4]; and eligible nonparticipants tend to have higher incomes and assets, have fewer children, and be less likely to have disabled members than SNAP participants [5].

SNAP participants receive benefits through an Electronic Benefit Transfer (EBT) card that can be used to purchase most foods and beverages (except alcohol, tobacco, dietary supplements, hot foods, and foods for on-premise consumption) at SNAP-authorized stores. There is overwhelming evidence that SNAP reduces food insecurity, which is the central goal of the program [6,7]. In recent years, there is a debate among policymakers and public health researchers and advocates about whether SNAP may be restructured to promote a healthier diet [8,9]. The discussion about SNAP can be informed by research on the association between SNAP and nutrition outcomes. Using nationally representative datasets, some observational studies found SNAP participants to have lower-quality diets or purchases overall, as indicated by a lower score on the Healthy Eating Index (HEI) or a modified Alternate HEI, than income-eligible nonparticipants [10–13]. Results at the food/nutrient group level or by demographic group are more mixed because of the multitude of dimensions in measurement, consumer heterogeneity and differences in analytical approaches and data. For example, several studies reported that SNAP was not correlated with intake of total fruit and vegetables for adults and children [12,14]; one found SNAP was associated with lower fruit intake for female participants [15]; and one study concluded that SNAP increased fruit consumption [13]. In terms of grain consumption, SNAP participants had a lower whole grain intake [12,16], compared with income-eligible nonparticipants, even though some studies showed that total grain consumption was almost the same [14,17,18]. The literature also shows mixed results on the intake of milk, meats, solid fats, beverages, and added sugars [12,13,18–21].

This study aims to examine the association of SNAP participation with the nutritional quality of food-at-home (FAH) purchases among low-income households and to investigate the heterogeneity among households with different nutrition attitudes. This study fills two gaps in the literature. First, few prior studies controlled for the influences of food prices on diet quality while examining the SNAP-diet relationship. Food prices are an important determinant of food choices and omitting food prices may create biases in the analysis. We controlled for food prices by including a household-specific price ratio in the model. This price ratio is calculated as the overall price of healthy (Guiding stars: 1 to 3 ratings) foods divided by the overall price of less healthy (0 ratings) foods. Second, previous studies revealed that positive attitudes towards healthy eating were associated with better diet quality, as measured by a higher HEI

score, lower dietary energy density, and higher consumption of fruit and vegetables [22–24]. However, few studies have investigated the heterogeneity in the association between SNAP and diet quality among consumers with different nutrition attitudes. Addressing this will inform the debate on the potential restructuring of SNAP.

# Methods

## Study sample

The USDA's National Household Food Acquisition and Purchase Survey (FoodAPS) is a nationally representative survey of household food purchases and acquisition. FoodAPS was fielded between April 2012 and January 2013 and 4,826 households completed the seven-day survey. The survey captures information on food items intended for consumption at home and away from home. Households were asked to scan barcodes on food, save receipts from stores and restaurants, and record information in provided food books. Our analysis focused on FAH purchases, including nine major food groups classified by USDA [25]: grain, vegetables, fruit, milk products, meat/beans, prepared meals/sides/salads, oils/fats/gravies, beverages, and sweet/salty snacks. "Oils/fats/gravies" food group includes fats, oils, salad dressings, gravies, sauces, condiments and spices, and "sweet/salty snacks" food group includes desserts, sweets, candies, and salty snacks.

FoodAPS includes SNAP households, low-income nonparticipants, and higher-income households. Throughout the entire analysis, we focused exclusively on low-income households with income below 185% FPL [26,27]. SNAP participation was verified by matching administrative records from state SNAP agencies and by tracking SNAP EBT card transactions using the Anti-fraud Locator EBT Retailer Transactions system. Our sample includes 2,218 low-income households, among whom 1,184 are SNAP participants and 1,034 are nonparticipants. The University of Georgia IRB determined that the proposed activity is not research involving human subjects as defined by DHHS and FDA regulations. We used public-use FoodAPS data and all data were fully anonymized before we accessed them.

## Measures

We used the Guiding Stars rating to measure the nutritional quality of each household's FAH purchases [28]. The Healthy Eating Index-2010 (HEI-2010) score was also used as an alternative measure to check the robustness of results.

Guiding stars is a nutrition guidance program that was first implemented at the Hannaford supermarket chain in September 2006 and is now available at several major grocery chains and thousands of food service facility locations across the United States. This program aims to translate nutrition facts into a simple rating that is easier for consumers to rank the healthfulness of food items. The Guiding Stars applies a nutrient profiling algorithm to nutrient density measures to rate the nutritional quality of food items on a 0–3 star rating [28]. First, the nutrients of a food item are scored to reflect dietary recommendations from authoritative scientific bodies (e.g., the *Dietary Guidelines for Americans (DGA)*) [28,29]. Health-promoting nutrients such as vitamins receive positive scores, and health-risking nutrients such as sodium receive negative scores. Second, food items with negative total scores are assigned a 0-star rating (e.g., candy), and this means that the food item does not meet the nutritional criteria to receive a star rating. Food items with positive total scores are classified into 1 star, 2 stars, and 3 stars to indicate good, better, and best nutrition value, respectively.

We calculated the gram-weighted average Guiding Stars rating of all items purchased by each household to measure the overall nutrition quality of its FAH purchases. Using grams to

weight food item-level Guiding Stars ratings prevents the household average star rating from being unduly influenced by energy-dense foods.

For robustness, we used the HEI-2010 score as an alternative to Guiding Stars rating to measure the nutritional quality of food purchases. The HEI-2010 uses a density approach to set standards, such as servings per 1,000 calories or as a percentage of calories [30]. The HEI-2010 ranges from 0 to 100 and is the sum of 12 component scores, each of which measures conformance to an aspect of the 2010 *DGA*. The scores increase with relative increases in dietary constituents that are encouraged such as fruit and decrease with relative increases in dietary constituents that are recommended in moderation such as added sugars. A higher HEI-2010 score represents a higher nutritional quality of food purchases.

### Statistical analysis

Descriptive statistics for the nutritional quality of FAH purchases and households' characteristics were generated for low-income households and stratified by SNAP status. The t-test and Chi-square tests were used to test the difference between SNAP households and nonparticipants. Multivariate regression was applied to examine the association of SNAP with the Guiding Stars rating of FAH purchases and to capture heterogeneity in the nutrition-SNAP association that can be tracked to a household's nutrition attitude. A household's nutrition attitude was measured using its response to a question on whether the household searched for online nutrition information in the last 2 months. Households that searched online for nutrition information were treated as *nutrition-oriented* households; those who did not were considered *less nutrition-oriented* households. For robustness, we also created an alternative measure of nutrition attitudes based on reported use of the nutrition facts label. The use of the nutrition facts label is based on the survey question "how often do you use Nutrition Facts panel", households that answered "always", "most of the time", and "sometimes" were treated as nutrition-oriented households, and households that answered "rarely" and "never" were treated as less nutrition-oriented households. We also used multivariate regressions to examine the associations of SNAP with energy acquired and nutrient density. Nutrient density was defined as the amount of the nutrient per 100 kcal of household's FAH purchases. The associations were examined for each of the following nutrients: saturated fat, sodium, cholesterol, added sugar, dietary fiber, whole grains, and vitamins/minerals.

Multivariate regressions were adjusted for the following covariates: cost of healthier foods, which is a household-specific ratio calculated as the overall price of starred (1 to 3-star) foods relative to the overall price of 0-star foods (Appendix A in S1 File for detailed information); household self-rated financial condition; house ownership status; household composition (household size, proportions of children, older adults, Hispanic race, obese members, smoker, and household members in poor health); household's meal planner's education; Special Supplemental Nutrition Program for Women, Infants, and Children (WIC) participation status; monthly food expenditure; and indicator variables for rurality of a household's location, food security status, and whether the household visited a food pantry/bank in the past month for groceries. Analyses accounted for survey design and sample weights and were conducted in 2019 using SAS (version 9.4) [31].

### Results

The average Guiding Stars rating of low-income households' FAH purchases was 0.664 points (out of a maximum of 3 points; Table 1). The Guiding Stars rating of SNAP participants was 0.558 points, 0.173 points ($p<0.001$) lower than the average rating of 0.731 points of nonparticipants. This suggested that SNAP households purchased foods with lower nutritional quality

**Table 1. Summary statistics of low-income households' characteristics.**

| Variable | Overall | SNAP | Income-eligible Non-SNAP | *P-value* |
|---|---|---|---|---|
| **Nutritional Quality** | | | | |
| Guiding Stars rating | 0.664 (0.016) | 0.558 (0.022) | 0.731 (0.025) | <0.001 |
| HEI-2010 score | 49.174 (0.505) | 46.927 (0.496) | 50.591 (0.725) | <0.001 |
| **Household Characteristics** | | | | |
| Household size (mean) | 2.522 (0.081) | 2.962 (0.088) | 2.245 (0.095) | <0.001 |
| Share with children (mean) | 0.181 (0.009) | 0.253 (0.014) | 0.135 (0.010) | <0.001 |
| Share with older adults (mean) | 0.226 (0.021) | 0.109 (0.019) | 0.301 (0.030) | <0.001 |
| Share with ≥1 Hispanic member (mean) | 0.209 (0.040) | 0.243 (0.051), | 0.188 (0.037) | 0.001 |
| Share with ≥1 obese member (mean) | 0.317 (0.010) | 0.371 (0.016) | 0.283 (0.015) | <0.001 |
| Share with ≥1 smoker (mean) | 0.256 (0.019) | 0.305 (0.020) | 0.224 (0.025) | <0.001 |
| Share with ≥1 member in poor health (mean) | 0.051 (0.006) | 0.072 (0.011) | 0.038 (0.007) | <0.001 |
| Food expenditure (mean, $/4 weeks) | 419.954 (12.070) | 500.652 (15.039) | 369.013 (13.530) | <0.001 |
| WIC participation (share) | 0.086 (0.008) | 0.149 (0.016) | 0.046 (0.008) | <0.001 |
| Poor financial condition (share) | 0.349 (0.015) | 0.248 (0.023) | 0.412 (0.019) | <0.001 |
| Own house (share) | 0.445 (0.028) | 0.316 (0.030) | 0.526 (0.034) | <0.001 |
| Visited food pantry/food bank (share) | 0.091 (0.009) | 0.146 (0.015) | 0.056 (0.009) | <0.001 |
| Rural (share) | 0.332 (0.049) | 0.304 (0.042) | 0.349 (0.058) | 0.022 |
| Food insecure (share) | 0.338 (0.015) | 0.432 (0.021) | 0.278 (0.018) | <0.001 |
| **Primary Respondent (PR) Characteristics** | | | | <0.001 |
| 10th grade or less (share) | 0.144 (0.016) | 0.167 (0.021) | 0.130 (0.016) | |
| 11th or 12th grade, no diploma (share) | 0.065 (0.011) | 0.088 (0.016) | 0.051 (0.010) | |
| High School diploma or GED (share) | 0.329 (0.016) | 0.338 (0.025) | 0.323 (0.020) | |
| College education (share) | 0.332 (0.017) | 0.316 (0.017) | 0.342 (0.025) | |
| Bachelor's degree (share) | 0.099 (0.010) | 0.076 (0.011) | 0.114 (0.014) | |
| Master's degree or more (share) | 0.028 (0.006) | 0.015 (0.006) | 0.037 (0.010) | |
| **Number of Households** | 2,218 | 1,184 | 1,034 | |

[a] Calculation of weighted means (standard errors in parentheses) accounts for the survey design of FoodAPS.

[b] The t-test and Chi-square test are used to test the difference between SNAP households and nonparticipants. Statistical significance is indicated by the p-value. A Chi-square test is used to test the overall difference in PR characteristics between SNAP and income-eligible non-SNAP households.

[c] Definition of variables: Food expenditure–household monthly food expenditures, including both FAH and food-away-from-home expenditures, calculated as 4 times the reported one-week expenditures; Household financial condition–household's self-rated financial condition is comfortable and secure; Food pantry/food bank—household went to a food bank or food pantry in past 30 days for groceries; Food insecure–the household is food insecure based on USDA's 30-day Adult Food Security Scale, households with adult members in low food security and very low food security are categorized as being "food insecure".

[d] Children are defined as age ≤18 years, an older adult is defined as age≥65 years.

than nonparticipants. We observed similar patterns in the comparison of HEI-2010 scores. The average HEI-2010 score of low-income households was 49.17 points, out of a maximum of 100 points. SNAP households had an average score of 46.93 points, 3.66 points ($p<0.001$) lower than nonparticipants.

SNAP households have different characteristics from non-SNAP households. Previous studies showed that compared to non-SNAP households, SNAP households have lower incomes and assets. Further, single-parent households, nonelderly households, and households with more children are more likely to participate in SNAP [5]. Our study observed similar patterns: compared to nonparticipants, SNAP households were larger, more likely to have children and Hispanic members, and less likely to have older adults. In terms of health status, SNAP households had larger proportions of obese members, smokers, and members in poor

**Table 2. Comparison of Guiding Stars rating and HEI-2010 score between SNAP and low-income non-SNAP households by nutrition attitude.**

|  | SNAP participants (1) | Income–eligible Non-SNAP participants (2) | Mean Difference (1)—(2) |
|---|---|---|---|
| **Guiding Stars rating** |  |  |  |
| Nutrition-oriented | 0.631 (0.038) | 0.698 (0.058) | -0.067 |
| Less nutrition-oriented | 0.538 (0.025) | 0.739 (0.025) | -0.200*** |
| **HEI-2010 score** |  |  |  |
| Nutrition-oriented | 50.413 (1.339) | 52.203 (1.636) | -1.790 |
| Less nutrition-oriented | 45.976 (0.574) | 50.250 (0.845) | -4.274*** |

[a] Calculation of weighted means (standard errors in parentheses) accounts for the survey design of FoodAPS.

[b] The *t*-test is used to test the difference between SNAP households and nonparticipants. Boldface indicates statistical significance (*** $p<0.01$, ** $p<0.05$, * $p<0.1$).

self-rated health status. SNAP households were also more likely to be food insecure—15 percentage points higher than nonparticipants, 43.2% vs. 27.8%. Also, SNAP households were more likely to participate in the WIC program and more than twice as likely to have visited a food pantry/bank in the past month for groceries than nonparticipants. Relative to nonparticipants, smaller proportions of SNAP households owned a house, were satisfied with their financial condition and had post-college education. Additionally, SNAP households spent $130/month more on food than nonparticipants, partly explained by the larger size of SNAP households.

Table 2 compares the nutritional quality of SNAP and non-SNAP households by nutrition attitude. Nutrition-oriented households were those that searched online nutrition information in the last 2 months, those who did not were treated as less nutrition-oriented households. Among the 2,218 low-income households, 471 were defined as nutrition-oriented households, and 1,747 were less nutrition-oriented households. The number of nutrition-oriented and less nutrition-oriented households by SNAP status was reported in S4 Table in S1 File. Compared to nonparticipants, SNAP participants on average had 0.2 points lower Guiding Stars rating than nonparticipants among less nutrition-oriented households, while the difference was not significant among nutrition-oriented households. Similarly, among less nutrition-oriented households, SNAP households had 4.274 points lower HEI-2010 score than nonparticipants. In comparison, among nutrition-oriented households, the difference in HEI-2010 score between SNAP and non-SNAP households was not significant.

Next, we assessed the association of SNAP with the household's Guiding Stars rating using multivariate regression to control for observed differences in household characteristics. The results are shown in Table 3. The interaction term *SNAP×NutritionSearch* captured the heterogeneity in the nutrition-SNAP association that can be tracked to a household's nutrition attitude. For less nutrition-oriented households, SNAP was associated with a statistically significant 0.097 points ($p = 0.018$, about 13.13%) lower Guiding Stars rating than nonparticipants. In comparison, for nutrition-oriented households, SNAP was associated with a statistically insignificant 0.008 points ($p = 0.890$) lower Guiding Star rating. The 0.008-point decline is calculated by summing the coefficients on SNAP participation ($-0.097$) and *SNAP×NutritionSearch* (0.089). We also found that a one-unit increase in the ratio of starred food price to unstarred food price was associated with a 0.085-point ($p = 0.027$) lower Guiding Stars rating. Households that went to a food pantry/bank in the past month for groceries were more likely to have a lower Guiding Stars rating and so were larger households. Households with children and smokers had lower Guiding Stars ratings, while households with at least one Hispanic member had higher Guiding Stars ratings.

**Table 3. Associations between nutritional quality and covariates among low-income households.**

| Variables | Guiding Stars rating | |
|---|---|---|
| | Coefficient | Std. Err. |
| SNAP participation (Yes = 1) | -0.097** | 0.039 |
| SNAP×NutritionSearch | 0.089 | 0.067 |
| NutritionSearch (Yes = 1) | -0.009 | 0.058 |
| WIC participation (Yes = 1) | 0.013 | 0.031 |
| Food price ratio | -0.085** | 0.037 |
| Food insecure (Yes = 1) | 0.016 | 0.041 |
| Standardized food expenditure | 0.017 | 0.013 |
| Rural (Yes = 1) | -0.050 | 0.042 |
| Household size | -0.023** | 0.011 |
| Share with children | -0.312*** | 0.060 |
| Share with older adults | 0.085 | 0.054 |
| Share with ≥1 Hispanic member | 0.125*** | 0.040 |
| Share with ≥1 obese member | -0.053 | 0.041 |
| Share with ≥1 smoker | -0.225*** | 0.047 |
| Share with ≥1 member in poor health | -0.014 | 0.074 |
| Household financial condition | 0.008 | 0.032 |
| Own house (Yes = 1) | 0.034 | 0.043 |
| Food pantry/food bank (Yes = 1) | -0.090** | 0.043 |
| Primary Respondent 's highest education | 0.017 | 0.013 |
| Constant | 0.837*** | 0.065 |
| N | 2,218 | |
| $R^2$ | 0.130 | |

[a]Multivariate regression is applied to examine the association of SNAP with the Guiding Stars rating of FAH purchases.

[b] The estimates use sample weights and control for survey design. Boldface indicates statistical significance (*** $p<0.01$, ** $p<0.05$, * $p<0.1$).

[c] *NutritionSearch* is an indicator variable for whether a household searched online nutrition information in the last 2 months. *SNAP×NutritionSearch* is the interaction between the SNAP participation variable and *NutritionSearch*.

We compared per capita energy and nutrients acquired or purchased per 100 kcal of FAH energy for SNAP households compared with income-eligible nonparticipants (Table 4). Compared to nonparticipants, SNAP households on average acquired more FAH energy per person (2710 kcal) over the one-week survey period. Also, SNAP participants acquired foods that were more dense in added sugars, but less dense in encouraged nutrients. Specifically, per 100 kcal of FAH energy, SNAP participants on average acquired 0.139 tsp. more added sugars (1.076 vs 0.937), but 0.007 g fewer vitamins/minerals (0.082 vs 0.089), 0.176 g less dietary fiber (0.662 vs 0.838), and 0.011 oz. less whole grains (0.033 vs 0.044).

Table 5 reported the association of SNAP with nutrient density and per capita FAH energy acquired using multivariate regressions. For nutrient densities, none of the *SNAP* coefficient estimates reached statistical significance. However, each *SNAP* coefficient took the expected sign consistent with the comparison of means in Table 4. Except for sodium, the nutrient density of FAH purchases by SNAP participants was lower than low-income nonparticipants for encouraged nutrients and higher for nutrients to limit. Except for the nutrient density regressions for dietary fiber and vitamins/minerals, all coefficients on the interaction term

**Table 4. Comparison of per capita energy acquired (kcal) and nutrients densities (per 100 kcal of FAH purchases) between SNAP and low-income non-SNAP households.**

| | SNAP participants (1) | Income–eligible Non-SNAP participants (2) | Mean Difference (1)—(2) |
|---|---|---|---|
| Per capita energy acquired (kcal/7-day) | 17300 (855.14) | 14590 (741.01) | 2710 ** |
| **Nutrients to limit** | | | |
| Saturated fat (g) | 1.351 (0.023) | 1.311 (0.037) | 0.040 |
| Cholesterol (g) | 0.013 (0.001) | 0.013 (0.0004) | 0 |
| Added sugars (tsp eq.) | 1.076 (0.038) | 0.937 (0.034) | 0.139** |
| Sodium (g) | 0.176 (0.007) | 0.209 (0.027) | -0.033 |
| **Nutrients to encourage** | | | |
| Vitamins/minerals (g) | 0.082 (0.001) | 0.089 (0.002) | -0.007** |
| Dietary fiber (g) | 0.662 (0.024) | 0.838 (0.034) | -0.176*** |
| Whole grain (oz eq.) | 0.033 (0.003) | 0.044 (0.003) | -0.011** |

[a] Calculation of weighted means (standard errors in parentheses) accounts for the survey design of FoodAPS.

[b] The t-test is used to test the difference between SNAP households and nonparticipants. Boldface indicates statistical significance (*** $p<0.01$, ** $p<0.05$, * $p<0.1$).

[c]. Vitamins/minerals here is the total purchased amount of vitamin A, vitamin B-6, vitamin B-12, vitamin C, vitamin D (D2+D3), vitamin E, vitamin K (phylloquinone), iron, thiamin, riboflavin, niacin, folic acid, phosphorus, magnesium, zinc, selenium, and copper.

*SNAP×NutritionSearch* had signs opposite to the corresponding coefficients on *SNAP*. This was consistent with the results in Table 3 where being nutrition-oriented counteracted the influence of SNAP on nutritional quality. For less nutrition-oriented households, SNAP participation was associated with higher per capita FAH energy acquired. A similar magnitude of this association was observed among nutrition-oriented households, although it was not statistically significant at conventional levels ($p = 0.126$). The positive influence of SNAP on FAH energy acquisition is expected because participants substitute FAH items for food away from home.

## Robustness checks

We re-estimated the association between SNAP and nutritional quality of households' FAH purchases using alternative specifications of regressions and alternative measures of nutritional quality and nutrition attitudes. The main result regarding the heterogeneity in SNAP's association with nutritional quality remained unchanged.

First, we examined the association of SNAP with households' Guiding Stars rating using separate regressions for the two types of households distinguished by online nutrition searches, i.e., one for nutrition-oriented households and the other for less nutrition-oriented households. SNAP was associated with 0.104 points ($p = 0.011$) lower Guiding Stars rating than nonparticipants among less nutrition-oriented households, while SNAP was not associated with differences in ratings among nutrition-oriented households (S1 Table in S1 File). Second, we used the HEI-2010 as an alternative measure of nutritional quality. Among less nutrition-oriented households, SNAP was associated with 2.106 points ($p = 0.056$) lower HEI-2010 score than nonparticipants. Again, there SNAP was not associated with HEI among nutrition-oriented households (S2 Table in S1 File). Third, we used households' responses about whether they used the nutrition facts label as an alternative measure of nutrition attitudes. Households that used nutrition facts labels were defined as nutrition-oriented households (N = 1,332), and others were less nutrition-oriented households (N = 886). The number of nutrition-oriented and less nutrition-oriented households by SNAP status based on this alternative definition of nutrition attitude was reported in S4 Table in S1 File. For less nutrition-oriented households,

**Table 5. Associations between covariates and nutrients densities (per 100 kcal of FAH purchases) and per capita energy acquired among low-income households.**

| Variables | Saturated fat (g) | Standardized Cholesterol | Added sugars (tsp eq.) | Sodium(g) | Dietary fiber (g) | Vitamins/ minerals (g) | Whole grain (oz.) | Standardized energy |
|---|---|---|---|---|---|---|---|---|
| SNAP participation (Yes = 1) | 0.034 (0.040) | 0.073 (0.071) | 0.036 (0.073) | -0.039 (0.033) | -0.083 (0.054) | -0.001 (0.003) | -0.008 (0.005) | 0.146*** (0.051) |
| SNAP*NutritionSearch | -0.031 (0.079) | -0.112 (0.168) | -0.062 (0.131) | 0.015 (0.047) | -0.018 (0.096) | -0.001 (0.006) | 0.011 (0.016) | 0.004 (0.098) |
| NutritionSearch (Yes = 1) | 0.004 (0.070) | 0.150 (0.090) | -0.004 (0.101) | -0.024 (0.041) | 0.034 (0.082) | 0.005 (0.004) | -0.004 (0.012) | -0.153** (0.068) |
| WIC participation (Yes = 1) | 0.041 (0.049) | 0.130 (0.094) | -0.076 (0.067) | 0.020 (0.024) | -0.010 (0.036) | 0.005** (0.002) | 0.002 (0.006) | -0.119* (0.069) |
| Food insecure (Yes = 1) | -0.088** (0.043) | -0.090 (0.062) | -0.032 (0.072) | -0.030 (0.025) | 0.029 (0.048) | 0.001 (0.002) | -0.003 (0.004) | -0.076* (0.045) |
| Standardized food expenditure | 0.022 (0.015) | 0.001 (0.028) | -0.020 (0.022) | 0.018 (0.012) | -0.021 (0.020) | -0.005*** (0.001) | -0.0001 (0.002) | 0.699*** (0.037) |
| Rural (Yes = 1) | 0.066* (0.033) | -0.012 (0.073) | 0.020 (0.062) | -0.045 (0.032) | -0.064 (0.046) | -0.001 (0.003) | 0.004 (0.007) | 0.051 (0.065) |
| Household size | -0.011 (0.013) | 0.002 (0.021) | 0.056*** (0.018) | -0.011 (0.007) | -0.019 (0.013) | -0.001 (0.001) | -0.001 (0.001) | -0.259*** (0.019) |
| Share with children | 0.074 (0.088) | -0.337** (0.137) | 0.274** (0.109) | -0.034 (0.035) | -0.196*** (0.070) | -0.009 (0.006) | -0.014 (0.010) | -0.032 (0.103) |
| Share with older adults | 0.046 (0.072) | -0.055 (0.088) | -0.070 (0.096) | 0.035 (0.035) | 0.071 (0.083) | -0.004 (0.006) | -0.018* (0.009) | 0.189** (0.088) |
| Share with ≥1 Hispanic member | -0.068 (0.048) | 0.099 (0.075) | -0.059 (0.078) | 0.021 (0.034) | 0.172*** (0.034) | 0.003 (0.003) | -0.007* (0.004) | -0.002 (0.067) |
| Share with ≥1 obese member | 0.115 (0.084) | -0.003 (0.091) | -0.123* (0.064) | 0.038 (0.035) | -0.069 (0.061) | -0.007 (0.004) | -0.008 (0.011) | 0.086 (0.071) |
| Share with ≥1 smoker | 0.118 (0.073) | 0.006 (0.107) | 0.226 (0.141) | 0.064 (0.053) | -0.213** (0.083) | -0.008 (0.006) | -0.030*** (0.008) | 0.156* (0.084) |
| Share with ≥1 member in poor health | -0.035 (0.090) | 0.041 (0.183) | 0.051 (0.115) | -0.065 (0.044) | 0.028 (0.092) | 0.0004 (0.004) | 0.012 (0.014) | -0.001 (0.127) |
| Household financial condition | -0.036 (0.070) | 0.101 (0.063) | -0.130** (0.052) | -0.036 (0.032) | -0.019 (0.047) | 0.007* (0.004) | 0.004 (0.006) | -0.145*** (0.043) |
| Own house (Yes = 1) | -0.005 (0.044) | -0.007 (0.064) | -0.043 (0.056) | 0.014 (0.032) | 0.087** (0.040) | 0.005** (0.003) | 0.006 (0.007) | 0.019 (0.047) |
| Food pantry/food bank (Yes = 1) | 0.108 (0.101) | 0.188 (0.128) | 0.017 (0.083) | 0.037 (0.040) | -0.050 (0.058) | 0.002 (0.004) | -0.002 (0.006) | 0.046 (0.069) |
| Primary Respondent 's highest education | 0.020 (0.016) | -0.027 (0.036) | -0.058*** (0.017) | -0.011 (0.009) | 0.028 (0.022) | 0.002 (0.001) | 0.002 (0.002) | -0.022 (0.017) |
| Constant | 1.218*** (0.101) | 0.085 (0.185) | 1.052*** (0.107) | 0.278*** (0.083) | 0.790*** (0.087) | 0.081*** (0.006) | 0.051*** (0.012) | 0.905*** (0.117) |
| N | 2,218 | 2,218 | 2,218 | 2,218 | 2,218 | 2,218 | 2,218 | 2,218 |
| $R^2$ | 0.024 | 0.018 | 0.060 | 0.013 | 0.064 | 0.054 | 0.035 | 0.384 |

[a] Multivariate regressions are used to examine the associations of SNAP with nutrients densities.

[b] Standard errors are in parentheses, and the estimates use sample weights and control for survey design. Boldface indicates statistical significance (*** $p < 0.01$, ** $p < 0.05$, * $p < 0.1$).

[C] Standardized cholesterol and energy are calculated by subtracting the mean from the original value and dividing the difference by the standard deviation.

[d]. Vitamins/minerals here is the total purchased amount of vitamin A, vitamin B-6, vitamin B-12, vitamin C, vitamin D (D2+D3), vitamin E, vitamin K (phylloquinone), iron, thiamin, riboflavin, niacin, folic acid, phosphorus, magnesium, zinc, selenium, and copper.

SNAP was associated with 0.120 points ($p = 0.027$) lower Guiding Stars rating compared with nonparticipants, but there was no association if SNAP with Guiding Stars ratings for nutrition-oriented households (S3 Table in S1 File).

There may be a concern with using food security status as an explanatory variable if SNAP causally alleviates food insecurity. Therefore, we also conducted a regression analysis without the food security indicator. The findings remain unchanged.

## Discussion

Many previous studies found SNAP participants to have lower diet quality than their income-eligible nonparticipating counterparts [11,13,15,32]. We found that SNAP participation was associated with the lower nutritional quality of FAH purchases among less nutrition-oriented households, but not among nutrition-oriented households. This heterogeneity in the SNAP-nutritional quality association may have important policy implications. For example, some researchers and public health advocates have proposed to restrict SNAP-eligible items to healthy foods [8,9,33,34], similar to the WIC program which prescribes only healthier food options. Opponents to these proposals have cited a possible stigma-induced reduction in SNAP enrollment, added administrative and retailer costs, and that implementing restrictions may make the program less attractive to low-income families, who rely on the flexibility of what can be purchased with SNAP to meet their food needs.

The merit of the SNAP restrictions is premised on the existence of a negative association between SNAP and nutritional quality. One study calculated the potential impact of restricting SNAP to exclude energy-dense foods and found that expenditures on the restricted items would decrease by $1.6 to $4.8 if $10 of SNAP benefits would have otherwise been spent [35]. Leveraging healthy diets could generate health benefits, such as preventing cardiovascular diseases and gain quality-adjusted life years among SNAP participants [36]. Our findings imply that SNAP restrictions might lead to improved nutritional outcomes and better health outcomes for less nutrition-oriented households. The lack of a SNAP-diet relationship for nutrition-oriented participants suggests that the intended benefit of the proposed changes may not reach this subgroup of SNAP population. It is even possible that low-income nutrition-oriented households become worse off if the restrictions reduce food security by discouraging participation in SNAP. Because food security is positively associated with health [37], there is a potential for SNAP restrictions to increase health disparity among low-income households, which is something policymakers hope to avoid. However, it is worth noting that there is not sufficient empirical evidence to suggest whether or not SNAP restrictions would work. One randomized controlled trial found SNAP restrictions alone would not lead to improved nutrition [38]. However, that study did not examine whether there was heterogeneity in the estimated effects among people with different nutrition attitudes.

SNAP Education (SNAP-Ed) is an optional component of SNAP that aims to increase the likelihood of healthy eating behaviors among the low-income population through direct nutrition education and social marketing. There is evidence that certain SNAP-Ed interventions are effective in promoting healthier behavioral and attitudinal changes for low-income children and adults [39–41]. This and the dependency of the SNAP-nutrition relationship on nutrition attitude underscores the promising role of successful SNAP-Ed interventions in closing the nutrition gap between less nutrition-oriented SNAP participants and low-income nonparticipants.

Among other policy-relevant results, the price of starred foods relative to unstarred foods was negatively associated with nutritional quality. This is consistent with the law of demand–a tenet of economics that predicts demand to increase in response to a decline in price [42]. As starred foods become more expensive relative to unstarred foods, the mix of purchase shifts toward unstarred foods and, hence, causes a reduction in nutritional quality. The USDA Food Insecurity Nutrition Incentive grant program is designed to support financial incentives that

reduce the relative price of fruit and vegetables for the SNAP population at farmers markets. There is mixed evidence on the program's effectiveness in increasing SNAP participants' fruit and vegetable intake [43–45]. Our result suggests that to improve overall nutritional quality, financial incentives may apply to a much broader range of healthy foods.

## Limitations

Our study has several limitations. First, our analysis is based on household purchase data, we cannot make definitive statements about food intake at the individual level because of food waste, stockpiling and intrahousehold sharing that make the linkage between household purchase and individual intake imperfect. Stockpiling is an issue because the one-week data collection of FoodAPS likely missed some nonperishable packaged foods (e.g., sugar-sweetened beverages) that were consumed but not purchased by some households in the survey week. For households who purchased during the survey, the purchased amount may be higher than consumption for storable food items. However, on average, the net effect of the two opposing forces may be small. Second, our analysis focuses on food-at-home purchases, therefore, our results cannot reflect the nutritional quality of total diet among low-income households. Third, although FoodAPS is the most recent nationally representative survey of food purchases and acquisitions that oversampled the low-income population, the data was collected in 2012. Future studies could examine the robustness of our results using other more recent datasets when becoming available. Fourth, our results are subject to potential omitted variable bias. Omitting unobserved factors that are both determinants of SNAP participation and nutritional quality of FAH purchases would create bias in the estimates on SNAP and its interaction with the indicator for nutrition attitude. However, to the extent that the same unobserved factors also influence household food security, financial condition, home ownership, and use of food pantry/bank, we can reduce the bias by including these variables as controls [46].

## Conclusions

SNAP offers millions of low-income Americans food benefits, while concern remains that the benefits may reduce the nutritional quality of participants' food basket. Our analysis found that for low-income households, SNAP was associated with lower nutritional quality of FAH purchases among less nutrition-oriented households, but not among nutrition-oriented households. This difference may have implications for the nutrition policies. For example, among less nutrition-oriented households, placing restrictions on using SNAP benefits to purchase unhealthy foods might lead to healthier eating and better health outcomes if participants do not increase unhealthy food purchases paid for by other incomes. However, the intended benefit of SNAP restrictions may not reach nutrition-oriented SNAP participants.

## Supporting information

**S1 File.**
(DOCX)

## Acknowledgments

We thank comments by Travis Smith, Greg Colson, Norbert Wilson, and session participants at 2019 Agricultural and Applied Economics Association Annual Meeting, and we have received permission from anyone named. We also thank two anonymous reviewers for their helpful comments on earlier versions of this paper.

## Author Contributions

**Conceptualization:** Biing-Hwan Lin, Chen Zhen.

**Formal analysis:** Yu Chen.

**Funding acquisition:** Chen Zhen.

**Methodology:** Yu Chen, Chen Zhen.

**Writing – original draft:** Yu Chen.

**Writing – review & editing:** Yu Chen, Biing-Hwan Lin, Lisa Mancino, Michele Ver Ploeg, Chen Zhen.

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
