## [Decision Letter · Decision Letter 0]

11 Aug 2020

PONE-D-20-17642

Nutritional Quality of Retail Food Purchases Is Not Associated with Participation in the Supplemental Nutrition Assistance Program for Nutrition-Oriented Households

PLOS ONE

Dear Dr. Zhen,

Thank you for submitting your manuscript to PLOS ONE. After careful consideration, we feel that it has merit but does not fully meet PLOS ONE’s publication criteria as it currently stands. Therefore, we invite you to submit a revised version of the manuscript that addresses the points raised during the review process.

We look forward to receiving your revised manuscript.

Kind regards,

Zhifeng Gao

Academic Editor

PLOS ONE

Journal Requirements:

2. Please refer to the specific statistical analyses performed as well as any post-hoc corrections to correct for multiple comparisons. If these were not performed please justify the reasons. Please refer to our statistical reporting guidelines for assistance (https://journals.plos.org/plosone/s/submission-guidelines.#loc-statistical-reporting).

3. In ethics statement in the manuscript and in the online submission form, please provide additional information about the patient records used in your retrospective study. Specifically, please ensure that you have discussed whether all data were fully anonymized before you accessed them and/or whether the IRB or ethics committee waived the requirement for informed consent. If patients provided informed written consent to have data from their medical records used in research, please include this information.

Reviewers' comments:

Reviewer's Responses to Questions

**Comments to the Author**

1. Is the manuscript technically sound, and do the data support the conclusions?

Reviewer #1: Partly

Reviewer #2: Yes

2. Has the statistical analysis been performed appropriately and rigorously? 

Reviewer #1: No

Reviewer #2: Yes

3. Have the authors made all data underlying the findings in their manuscript fully available?

Reviewer #1: Yes

Reviewer #2: Yes

4. Is the manuscript presented in an intelligible fashion and written in standard English?

Reviewer #1: Yes

Reviewer #2: Yes

5. Review Comments to the Author

Reviewer #1: By mainly use of multiple regression analysis s seen on lines 110 to 138, the investigators concluded that SNAP participation was associated with lower nutritional quality of food purchases among less nutrition-oriented households, but not among nutrition-oriented households. The lack of such an association for nutrition-oriented participants suggests that the intended nutrition benefit of placing restrictions on purchasing unhealthy foods with SNAP benefits may not reach this subgroup of SNAP population. The statistical applications are simple. However, the presentation of the results is very confusing. Specifically:

1. The results are presented by low income household and the text also discusses results by nutritional attitude and then performs a robustness analysis and demonstrates the consistency of the results. The tables in the text are mostly results by low income households and Table 2 appears to attempt to show the results by less nutritional vs. nutritional oriented household by manipulation of the regression coefficients. Is one to assume that all the households in the study are low income?

2. Except for presentation of the covariates in the multiple regression format there are no tables in the paper showing a side by side comparison of the less nutrition vs. nutrition oriented comparisons which would be of interest. Also, Table 2 with the manipulation of the -0.097+0.089 on line 176 can be confusing to a non statistically oriented reader. Also what is SNAP*Nutrition Search? Is that an interaction term and how does it differ from Nutrition Search (Yes=1)?

3. How much of the variation in the dependent measures are being explained by the regressions? There are no R-squares in the results?

Reviewer #2: This article addresses an important issue (is the intended restrictions on purchasing unhealthy foods with SNAP of nutrition benefit?), however this paper only partially answers this. While the paper is comprehensive, I found at times I had to carefully reread sections to understand the paper fully. There are some points that need to be clarified (stated below).

Abstract:

FoodAPS - State this in full.

16: among

Income-eligible nonparticipants. As someone not familiar with the details of SNAP, it would be useful to explain the income criteria for SNAP and describe the income-eligible non-participants. Why don't they receive SNAP? Are they different to the SNAP participants (partly addressed in results)?

In-text citations: Be consistent with space before the citation

Introduction:

55: It is not clear to me how this study controls for the influences of food prices? This may be stated in the paper but there is likely to be other readers who require this to be clearer.

60: research studies

Methods:

68: Is the age of the dataset a limitation? Is there no recent data?

74: "Oils ....

90: Please explain more about the Guiding Stars. Is this a common nutrient profiling tool in the US? What increments are used? Is there a commonly accepted cut-off for unhealthy and less healthy?

118: You use households that had online nutrition search as a key measure. Has this been used elsewhere? It is good that an alternative measure is also used but has either been verified against a comprehensive tool to assess nutrition attitude?

Results:

141: Add that the differences were statistically significant

156: As SNAP households seem to have different characteristics to non-SNAP, is it these differences that explain the differences observed in the study? You may have stated this in the results but it needs to be clearly addressed.

Nutrient density has been used as a measure. Can you also assess overall energy intake? This is obviously key to obesity.

The results do not present actual numbers of participants anywhere. It would be useful to know how many households were considered 'nutrition-oriented' for SNAP and non-SNAP participants by both methods.

The robustness check is useful. How many participants were considered 'nutrition-oriented' by both methods? How many participants had low HEI scores and low star scores?

Discussion

282: Replace much with many

How will the proposed change affect the less nutrition oriented households? The discussion addresses the impact on nutrition-oriented households only. This needs to be discussed further in the discussion and conclusion.

References:

Check that the article titles are in lower case.

6. PLOS authors have the option to publish the peer review history of their article (what does this mean?). If published, this will include your full peer review and any attached files.

Reviewer #1: No

Reviewer #2: No

---

## [Author Response · Author response to Decision Letter 0]

11 Sep 2020

Our full response is in the file named Response to Reviewers.

---

## [Decision Letter · Decision Letter 1]

23 Sep 2020

Nutritional Quality of Retail Food Purchases Is Not Associated with Participation in the Supplemental Nutrition Assistance Program for Nutrition-Oriented Households

PONE-D-20-17642R1

Dear Dr. Zhen,

We’re pleased to inform you that your manuscript has been judged scientifically suitable for publication and will be formally accepted for publication once it meets all outstanding technical requirements.

Kind regards,

Zhifeng Gao

Academic Editor

PLOS ONE

Additional Editor Comments (optional):

Reviewers' comments:

Reviewer's Responses to Questions

**Comments to the Author**

1. If the authors have adequately addressed your comments raised in a previous round of review and you feel that this manuscript is now acceptable for publication, you may indicate that here to bypass the “Comments to the Author” section, enter your conflict of interest statement in the “Confidential to Editor” section, and submit your "Accept" recommendation.

Reviewer #1: All comments have been addressed

Reviewer #2: (No Response)

2. Is the manuscript technically sound, and do the data support the conclusions?

Reviewer #1: Yes

Reviewer #2: (No Response)

3. Has the statistical analysis been performed appropriately and rigorously? 

Reviewer #1: Yes

Reviewer #2: (No Response)

4. Have the authors made all data underlying the findings in their manuscript fully available?

Reviewer #1: Yes

Reviewer #2: (No Response)

5. Is the manuscript presented in an intelligible fashion and written in standard English?

Reviewer #1: Yes

Reviewer #2: (No Response)

6. Review Comments to the Author

Reviewer #1: (No Response)

Reviewer #2: (No Response)

7. PLOS authors have the option to publish the peer review history of their article (what does this mean?). If published, this will include your full peer review and any attached files.

Reviewer #1: No

Reviewer #2: No

---

## [Editor Report · Acceptance letter]

7 Dec 2020

PONE-D-20-17642R1 

Nutritional quality of retail food purchases is not associated with participation in the Supplemental Nutrition Assistance Program for nutrition-oriented households 

Dear Dr. Zhen:

I'm pleased to inform you that your manuscript has been deemed suitable for publication in PLOS ONE. Congratulations! Your manuscript is now with our production department. 

Kind regards, 

on behalf of

Dr. Zhifeng Gao 

Academic Editor

PLOS ONE